

# Macrel: antimicrobial peptide screening in genomes and metagenomes

Célio Dias Santos-Júnior[1,2], Shaojun Pan[1,2], Xing-Ming Zhao[1,2] and
Luis Pedro Coelho[1,2]

[1] Institute of Science and Technology for Brain-Inspired Intelligence, Fudan University, Shanghai,
China
[2] Ministry of Education, Key Laboratory of Computational Neuroscience and Brain-Inspired
Intelligence, Shanghai, China

## ABSTRACT

**Motivation:** Antimicrobial peptides (AMPs) have the potential to tackle
multidrug-resistant pathogens in both clinical and non-clinical contexts. The recent
growth in the availability of genomes and metagenomes provides an opportunity for
in silico prediction of novel AMP molecules. However, due to the small size of
these peptides, standard gene prospection methods cannot be applied in this domain
and alternative approaches are necessary. In particular, standard gene prediction
methods have low precision for short peptides, and functional classification by
homology results in low recall.

**Results:** Here, we present Macrel (for metagenomic AMP classification and
retrieval), which is an end-to-end pipeline for the prospection of high-quality AMP
candidates from (meta)genomes. For this, we introduce a novel set of 22 peptide
features. These were used to build classifiers which perform similarly to the
state-of-the-art in the prediction of both antimicrobial and hemolytic activity of
peptides, but with enhanced precision (using standard benchmarks as well as a
stricter testing regime). We demonstrate that Macrel recovers high-quality AMP
candidates using realistic simulations and real data.

**Availability:** Macrel is implemented in Python 3. It is available as open source at
https://github.com/BigDataBiology/macrel and through bioconda. Classification of
peptides or prediction of AMPs in contigs can also be performed on the webserver:
https://big-data-biology.org/software/macrel.

**Subjects** Bioinformatics, Computational Biology, Ecosystem Science, Microbiology, Data Mining
and Machine Learning
**Keywords** Antimicrobial peptides, Metagenomes, Bioprospection, Machine learning, Microbiome,
Genomes

## INTRODUCTION

Antimicrobial peptides (AMPs) are short proteins (containing fewer than 100 amino
acids) that can decrease or inhibit bacterial growth. Given the dearth of novel antibiotics in
recent decades and the rise of antimicrobial resistance, prospecting naturally-occurring
AMPs is a potentially valuable source of new antimicrobial molecules (*Theuretzbacher
et al., 2019*). The increasing number of publicly available metagenomes and
metatranscriptomes revealed a multitude of microorganisms so far unknown, harboring

Corresponding author
Luis Pedro Coelho,
luis@luispedro.org

immense biotechnological potential (*Pascoal, Magalhães & Costa, 2020*; *Bernard et al., 2018*). This presents an opportunity to use these (meta)genomic data to find novel AMP sequences. However, methods that have been successful in prospecting other microbial functionality cannot be directly applied to small genes (*Saghatelian & Couso, 2015*), such as AMPs. In particular, there are two major computational challenges: the prediction of small genes in DNA sequences (either genomic or metagenomic contigs) and the prediction of AMP activity for small genes using homology-based methods.

Current automated gene prediction methods typically exclude small open reading frames (smORFs) (*Miravet-Verde et al., 2019*), as the naïve use of the methods that work for larger sequences leads to unacceptably high rates of false positives when extended to short sequences (*Hyatt et al., 2010*). A few recent large-scale smORFs surveys have, nonetheless, shown that these methods can be employed if the results are subsequently analyzed to eliminate spurious gene predictions. These procedures reveal biologically active prokaryotic smORFs across a range of functions (*Miravet-Verde et al., 2019*; *Sberro et al., 2019*).

Similarly, the prediction of AMP activity requires different techniques than the homology-based methods that are applicable for longer proteins (*Huerta-Cepas et al., 2017*). In this context, several machine learning-based methods have demonstrated high accuracy in predicting antimicrobial activity in peptides, when tested on curated benchmarks (*Xiao et al., 2013*; *Meher et al., 2017*; *Lata, Mishra & Raghava, 2010*; *Thakur, Qureshi & Kumar, 2012*; *Sharma et al., 2016*; *Bhadra et al., 2018*). However, to be applicable to the task of extracting AMPs from genomic data, an AMP classifier needs to be robust to gene mispredictions and needs to be benchmarked in that context. In particular, realistic evaluations need to reflect the fact that most predicted genes are unlikely to have antimicrobial properties.

Different AMP prediction methods employed alternative ways of representing the sequential, compositional, and physicochemical properties of peptide sequences to create either binary (AMP vs. non-AMP) or multi-class (e.g., antibacterial, antifungal…) classifiers (*Spänig & Heider, 2019*). The Collection of Antimicrobial Peptides website (CAMP R3) contains a selection of AMP prediction tools based on random forests (RF), support vector machines (SVM), artificial neural networks (ANN), and discriminant analysis (DA) trained on 257 features (*Waghu et al., 2016*). *Xiao et al. (2013)* presented iAMP-2L, which uses a fuzzy K-nearest neighbor algorithm and the pseudo–amino acid composition of AMPs, resulting in 46 features. Another multi-class AMPs predictor is the SVM-based iAMPpred, which similarly uses features representing physicochemical and structural properties of AMPs (*Meher et al., 2017*). In both systems, the same sequence may be identified as simultaneously belonging to different subclasses (e.g., both antibacterial and antifungal) (*Lin et al., 2019*). Later, *Bhadra et al. (2018)* proposed the use of distribution patterns of amino acid properties as features for a highly accurate RF classifier (AmPEP). *Veltri, Kamath & Shehu (2018)* introduced a deep learning method for AMP prediction using neural network models with convolutional layers and the amino acid sequence as a predictive feature, the AMP Scanner.

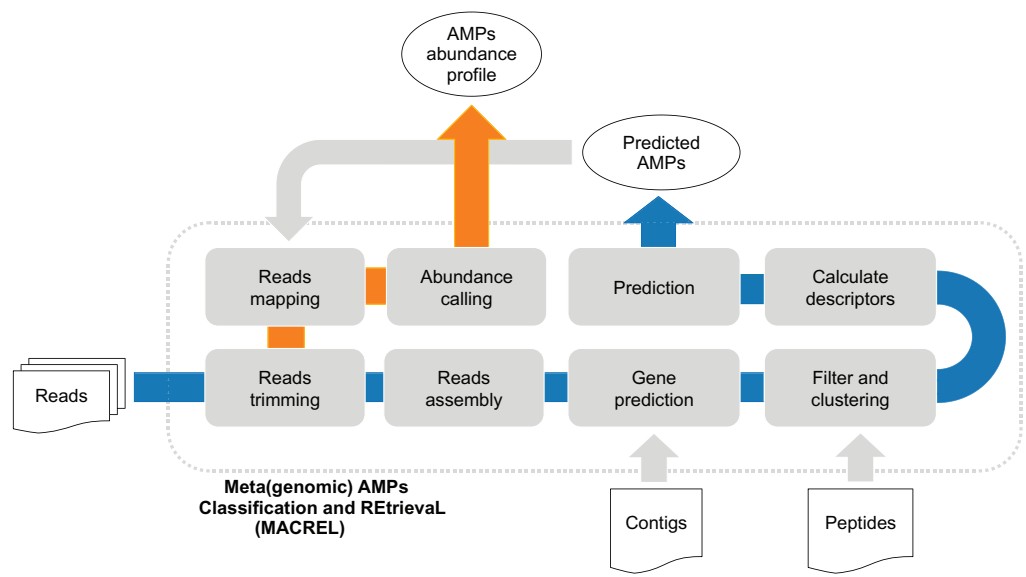

**Figure 1** **(Meta)genomic AMPs Classification and REtrievaL: Macrel pipeline.** The blue arrows show the Macrel workflow from the processing of reads until AMP prediction. The user can alternatively provide contigs or peptide sequences directly, skipping the initial steps of the short read pipeline. The orange arrow shows the abundance profiling of AMPs using Macrel output and reads.

These classifiers work on peptide sequences and are not directly applicable to microbial genomes or metagenomes. For this, we present Macrel—(Meta)genomic AMP Classification and Retrieval system—a pipeline that processes peptides, contigs, or reads from genomes and metagenomes, predicting AMP sequences (Fig. 1). Macrel is also capable of providing abundances profiles of a given set of AMPs in metagenomes. Unlike the systems described above, Macrel was trained with a very low proportion of AMPs to non-AMP peptides, simulating the conditions found in genomes and metagenomes, where only a small fraction of peptides will have antimicrobial activity. Furthermore, for applications to (meta)genomic data, the class imbalance in real data implies that high specificity is a more important metric than sensitivity.

# METHODS

## Macrel classifiers

### Features

Local features, those dependent on the order of the peptide sequence, were inspired by the composition-transition-distribution (CTD) framework *Dubchak et al. (1995, 1999)*. Physicochemical properties of a peptide at its N terminal are informative for the prediction of its antimicrobial activity (*Bahar & Ren, 2013*; *Bhadra et al., 2018*). Therefore, we defined features based on the normalized position of the first amino acid in a group of interest.

Global features, which are independent of amino acids primary sequence, were chosen to capture well-described AMP characteristics, such as the typical AMPs composition of approximately 50% hydrophobic residues, usual positive charge and folding into

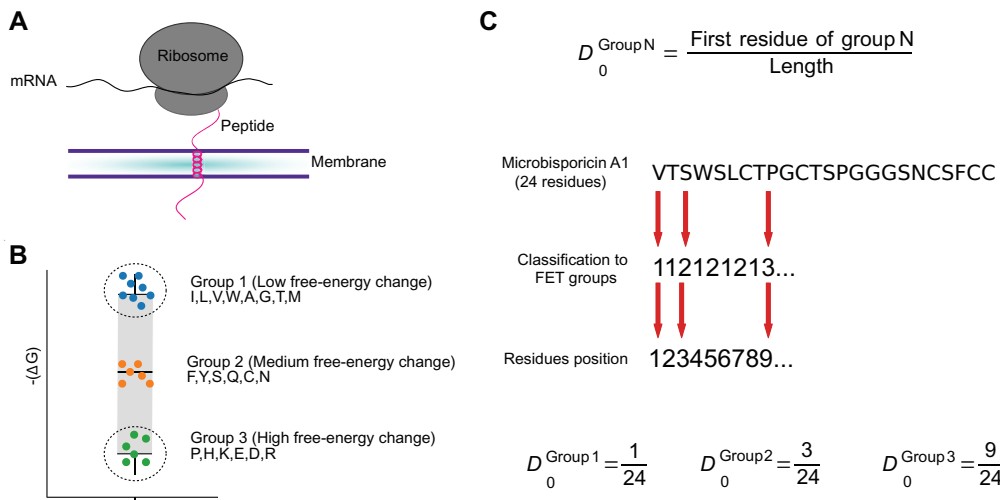

**Figure 2 The FET–Free energy transition—measures estimates the propensity of peptides to fold when transferring from water to the membrane.** The estimated change in the free-energy of the conformational change of an amino acid from random to organized structures in lipid membranes (A) was used to cluster amino acids into three groups (B). These groups were used to encode peptide sequences as the relative position of the first amino acid in each group (C).

amphiphilic ordered structures (*Zhang & Gallo, 2016*). The mechanism of antimicrobial activity also was summarized in Macrel's features by global descriptors of stability, amphiphilicity and predisposition of a peptide to bind to membranes.

Therefore, Macrel combines 6 local and 16 global features (see Table S1), grouped as:

1. A new local feature group (3 local features), defined as the relative position of the first occurrence of residues in three groups of amino acids defined by their free energy of transition in a peptide from a random coil in aqueous environment to an organized helical structure in a lipid phase *FET*—see Fig. 2. The groups are: (1, lowest FET): *ILVWAMGT*, (2, intermediate): *FYSQCN*, (3, highest): *PHKEDR* (*Von Heijne & Blomberg, 1979*).

2. Solvent Accessibility (3 local features), obtained by the distribution at first occurrence of residues organized in groups by solvent accessibility as described by *Bhadra et al. (2018)*, using the groups: (1, buried): *ALFCGIVW*, (2, exposed): *RKQEND*, and (3, intermediate): *MSPTHY*.

3. Amino acid composition (9 global features) as the fraction of amino acids in groups defined by their size (area/volume), polarity, charge and R-groups: acidic, basic, polar, non-polar, aliphatic, aromatic, charged, small, tiny (*Jhong et al., 2019*; *Nagarajan et al., 2019*).

4. Charge and solubility (2 global features): peptide charge (*Ebenhan et al., 2014*; *Chung et al., 2020*) and isoelectric point (*Fan et al., 2016*; *Wenzel et al., 2014*; *Chung et al., 2020*).

5. Indexes for multiple purposes (3 global features): instability, aliphaticity, propensity to bind to membranes (Boman (*Jhong et al., 2019*; *Chung et al., 2020*; *Boman, 2003*)).

6. Hydrophobicity (2 global features): hydrophobicity (KyteDoolittle scale) and hydrophobic moment at 100° to capture the helix momentum (*Ebenhan et al., 2014*; *Dathe et al., 1997*).

### Macrel prediction models

For AMP prediction, our training set is adapted from the one presented by *Bhadra et al. (2018)* by eliminating redundant sequences. The resulting set contains 3,268 AMPs (from diverse databases, most bench-validated) and 165,138 non-AMPs (a ratio of approximately 1:50). A random forest classifier with 101 tree was trained using scikit-learn (*Pedregosa et al., 2011*) (all parameters, except the number of trees, were set to their default values).

The hemolytic activity classifier was built similarly to AMP classifier. For this, we used the training set HemoPI-1 from *Chaudhary et al. (2016)*, which contains 442 hemolytic and 442 non-hemolytic peptides.

The datasets used in Macrel are extensively documented elsewhere (*Bhadra et al., 2018*; *Xiao et al., 2013*; *Veltri, Kamath & Shehu, 2018*; *Chaudhary et al., 2016*) and their description is available in the Table S2. Briefly, the AMP dataset is formed by unique sequences collected from ADP3, CAMPR3, LAMP databases. Non-AMP sequences were retrieved from the Uniprot database which were not annotated as AMP, membrane, toxic, secretory, defensing, antibiotic, anticancer, antiviral and antifungal. Hemolytic peptides dataset is composed of experimentally validated hemolytic peptides from Hemolytik database and randomly generated peptides from SwissProt as negative examples. No peptides containing non-canonical amino acids were kept.

### Prediction in genomes and metagenomes

For processing either genomes or metagenomes, Macrel (see Fig. 1) accepts as inputs paired-end or single-end reads in (possibly compressed) FastQ format and performs quality-based trimming with NGLess (*Coelho et al., 2019*). After this initial stage, Macrel assembles contigs using MEGAHIT (*Li et al., 2016*) (a minimum contig length of 1,000 base pairs is used). Alternatively, if available, contigs can be passed directly to Macrel.

Genes are predicted on these contigs with a modified version of Prodigal (*Hyatt et al., 2010*), which predicts genes with a minimal length of 30 base pairs (compared to 90 base pairs in the standard Prodigal release). The original threshold was intended to minimize false positives (*Hyatt et al., 2010*), as gene prediction methods, in general, generate more false positives in shorter sequences (smORFs) (*Höps, Jeffryes & Bateman, 2018*). *Sberro et al. (2019)* showed that reducing the length threshold without further filtering could lead to as many as 61.2% of predicted smORFs being false positives. In Macrel, this filtering consists of outputting only those smORFs (10–100 amino acids) classified as AMPs.

For convenience, duplicated sequences can be clustered and output as a single entity. For calculating AMP abundance profiles, Macrel uses Paladin (*Westbrook et al., 2017*) and NGLess (*Coelho et al., 2019*).

Protein synthesis in prokaryotes is started by N-formylmethionine (*Wingfield, 2017*). Post synthesis, circa 80–50% of the proteins undergo N-methionine excision (*Matheson, Yaguchi & Visentin, 1975*; *Waller, 1963*; *Giglione, Boularot & Meinnel, 2004*) so that this initial residue is not present in the active form of the peptide (*Giglione, Boularot & Meinnel, 2004*). As there is no tool to predict which peptides will undergo this process, we have chosen to always disregard an initial methionine when computing features, thus simulating the excision process.

## Benchmarking

### Methods to be compared

We compared the Macrel AMP classifier to the webserver versions of the following methods: CAMPR3 (including all algorithms) (*Waghu et al., 2016*), iAMP-2L (*Xiao et al., 2013*), AMAP (*Gull, Shamim & Minhas, 2019*), iAMPpred (*Meher et al., 2017*) and Antimicrobial Peptides Scanner v2 (*Veltri, Kamath & Shehu, 2018*). Results from AmPEP on this benchmark were obtained from the original publication (*Bhadra et al., 2018*). For all these comparisons, we used the benchmark dataset from *Xiao et al. (2013)*, which contains 920 AMPs and 920 non-AMPs.

The datasets from (*Xiao et al., 2013*) do not overlap. However, the training set used in Macrel and the test set from *Xiao et al. (2013)* do overlap extensively. Therefore, for testing, after the elimination of identical sequences, we used the out-of-bag estimate for any sequences that were present in the training set. Furthermore, as described below, we also tested using an approach which avoids homologous sequences being present in both the testing and training (see Table S2).

The benchmarking of the hemolytic peptides classifier was performed using the HemoPI-1 benchmark dataset formed by 110 hemolytic proteins and 110 non-hemolytic proteins previously established by *Chaudhary et al. (2016)*. Macrel model performance was compared against models created using different algorithms (*Chaudhary et al., 2016*): Support vector machines—SVM, K-Nearest Neighbor (IBK), Neural networks (Multilayer Perceptron), Logistic regression, Decision trees (J48) and RF. There is no overlap between the training set and the testing set for the benchmark of hemolytic peptides.

### Homology-aware benchmarking

Cd-hit (v4.8.1) (*Fu et al., 2012*) was used to cluster all sequences at 80% of identity and 90% of coverage of the shorter sequence. Only a single representative sequence from each cluster composed the dataset randomly split into training and testing partitions. The testing set was composed of 500 AMPs:500 non-AMPs. The training set contained 1,197 AMPs and was randomly selected to contain non-AMPs at different proportions (1:1, 1:5, 1:10, 1:20, 1:30, 1:40, and 1:50).

Using the training and testing sets, we tested four different methodologies: homology search, Macrel, iAMP-2L (*Xiao et al., 2013*) and AMP Scanner v.2 (*Veltri, Kamath & Shehu, 2018*) (these are the tools which enable users to retrain their classifiers). Homology search used blastp (*Camacho et al., 2009*), with a maximum $e$-value of 1e−5, minimum

identity of 50%, word size of 5, 90% of query coverage, window size of 10 and subject besthit option. Sequences lacking homology were considered misclassified.

### Benchmarks on simulated and real data

To test the Macrel short reads pipeline, 6 metagenomes were simulated at 3 different sequencing depths (40, 60 and 80 million reads of 150 bp) with ART Illumina v2.5.8 (*Huang et al., 2012*) using the pre-built sequencing error profile for the HiSeq 2500 sequencer. To ensure realism, the simulated metagenomes contained species abundances estimated from real human gut microbial communities (*Coelho et al., 2019*).

We processed both the simulated metagenomes and the isolate genomes used to build the metagenomes with Macrel to verify whether the same AMP candidates could be retrieved and whether the metagenomic processing introduced false positive sequences not present in the original genomes.

The 182 metagenomes and 36 metatranscriptomes used for benchmarking were published by *Heintz-Buschart et al. (2016)* and are available from the European Nucleotide Archive (accession number PRJNA289586). Macrel was used to process metagenome reads (see Table S3), and to generate the abundance profiles from the mapping of AMP candidates back to the metatranscriptomes. The results were transformed from counts to reads per million of transcripts.

### Detection of spurious sequences

To test whether spurious smORFs still appeared in Macrel results, we used Spurio (*Höps, Jeffryes & Bateman, 2018*) and considered a prediction spurious if the score was greater or equal to 0.8.

To identify putative gene fragments, the AMP sequences predicted with Macrel were validated through homology-searching against the non-redundant NCBI database (https://www.ncbi.nlm.nih.gov/). Predicted AMPs annotation was done by homology against the DRAMP database (*Fan et al., 2016*), which comprises circa 20k AMPs. The above-mentioned databases were searched with the blastp algorithm (*Camacho et al., 2009*), using a maximum $e$-value of $1 \times 10^{-5}$ and a word size of 3. Hits with a minimum of 70% of identity and 95% query coverage were kept and parsed to the best-hits after ranking them by score, $e$-value, identity, and coverage. To check whether the AMPs predicted by the Macrel pipeline were gene fragments, patented peptides or known AMPs, the alignments were manually evaluated.

### Implementation and availability

Macrel is implemented in Python 3 and R (*R Core Team, 2018*). Peptides (*Osorio, Rondón-Villarreal & Torres, 2015*) is used for computing features, and the classification is performed with scikit-learn (*Pedregosa et al., 2011*). For ease of installation, we made available a bioconda package (*Grüning et al., 2018*). The source code for Macrel is archived at DOI 10.5281/zenodo.3608055 (with the specific version tested in this manuscript being available as DOI 10.5281/zenodo.3712125).
The complete set of scripts used to benchmark Macrel is available at https://github.com/BigDataBiology/macrel2020benchmark and the newly simulated generated dataset of different sequencing depths is available at Zenodo (DOI 10.5281/zenodo.3529860).

# RESULTS

## Macrel: (Meta)genomic AMPs classification and REtrievaL

Here, we present Macrel (for *(Meta)genomic AMPs Classification and REtrievaL*, see Fig. 1), a simple, yet accurate, pipeline that processes either genomes or metagenomes/metatranscriptomes and predicts AMP sequences. We test Macrel with standard benchmarks for AMP prediction as well as both simulated and real sequencing data to show that, even in the presence of large numbers of (potentially artifactual) input smORFs, Macrel still outputs only a small number of high-quality candidates.

Macrel can process metagenomes (in the form of short reads), (meta)genomic contigs, or peptides. If short reads are given as input, Macrel will preprocess and assemble them into larger contigs. Automated gene prediction then extracts smORFs from these contigs which are classified into AMPs or rejected from further processing (see Fig. 1 and "Methods"). Putative AMPs are further classified into hemolytic or non-hemolytic. Unlike other pipelines (*Jhong et al., 2019*), Macrel can not only quantify known sequences, but also discover novel AMPs.

Macrel is also available as a webserver at https://big-data-biology.org/software/macrel, which accepts both peptides and contig sequences, and retrieves AMPs coded by their own genes.

## Novel set of protein descriptors for AMP identification

Two binary classifiers are used in Macrel: one predicts AMP activity and another the hemolytic activity (which is invoked only for putative AMPs). These are feature-based classifiers and use a set of 22 variables that capture the amphipathic nature of AMPs and their propensity to form transmembrane helices (see Table S1).

Peptide sequences can be characterized using local or global features: local features depend on the order of the amino acids, while global ones do not. Local features have been shown to be more informative when predicting AMP activity and its targets, while global features are more informative when predicting the potency of a given AMP (*Bhadra et al., 2018*; *Fjell et al., 2009*; *Boone et al., 2018*). Thus, Macrel combines both, including 6 local and 16 global features (see "Methods" and Table S1):

- *Free energy transition (FET)* (3 local features). This is a novel feature group, which was designed to capture the fact that AMPs usually fold from random coils in the polar phase to well-organized structures in lipid membranes (*Nagarajan et al., 2019*). Each amino acid is assigned to one of three groups of increasing free-energy change (*Von Heijne & Blomberg, 1979*). The three features consist of the position of the first amino acid in each group, normalized to the length of the sequence (see Fig. 2C). Earlier works had shown that the N-terminal is particularly informative for determining AMP

activity (*Bahar & Ren, 2013*; *Bhadra et al., 2018*). We adopted the fractional position encoding from the more general CTD framework (*Dubchak et al., 1995*, *1999*).

- *Solvent accessibility* (3 local features). Computed in the same way as the FET features, with amino acids groups representing levels of solvent accessibility.
- *Amino acid composition* (9 global features). As AMPs usually have biased amino acids composition (*Nagarajan et al., 2019*; *Jhong et al., 2019*), we used the fraction of amino acids falling into nine partially overlapping classes defined by charge, size, polarity, and hydrophobicity (see "Methods" and Table S1).
- *Charge* (1 global feature). AMPs typically contain approximately 50% hydrophobic residues (*Zhang & Gallo, 2016*; *Malmsten, 2014*; *Pasupuleti, Schmidtchen & Malmsten, 2012*), and their net charges are crucial to promote the peptide-induced membrane disruption (*Malmsten, 2014*; *Pasupuleti, Schmidtchen & Malmsten, 2012*; *Ringstad, Schmidtchen & Malmsten, 2006*).
- *Membrane binding and solubility in different media* (6 global features). These capture predisposition of peptides bind to membranes, and their solubility (*Ebenhan et al., 2014*; *Dathe et al., 1997*; *Jhong et al., 2019*).

All 22 descriptors used in Macrel are important for classification (see Fig. S1A). The fraction of acidic residues, charge, and isoelectric point were the most important variables in the hemolytic peptides classifier. Those variables tend to capture the electrostatic interaction between peptides and membranes, a key step in hemolysis. For AMP prediction, charge and the distribution parameters using FET and solvent accessibility are the most important variables. This is consistent with reports that cationic peptides (e.g., lysine-rich) show increased AMP activity (*Bhadra et al., 2018*; *Jhong et al., 2019*; *Nagarajan et al., 2019*).

## Compared to other tools, Macrel achieves the highest specificity, albeit at lower sensitivity

To evaluate the feature set and the classifier used in the context of the pipeline as a whole, we benchmark both the classifier implemented in Macrel, built with the training set adapted from *Bhadra et al. (2018)* (see "Macrel Prediction Models"), which consists of 1 AMP for each 50 negative examples, and a second AMP classifier (denoted Macrel[X]), which was built using the same features and methods, but using the training set from *Xiao et al. (2013)*, which contains 770 AMPs and 2,405 non-AMPs (approximately 1:3 ratio).

Benchmark results show that the AMP classifier trained with a more balanced dataset performs better than most of alternatives considered on this balanced benchmark, with AmPEP (*Bhadra et al., 2018*) achieving the best results (see Table 1).

In terms of overall accuracy on this benchmark, the AMP classifier implemented in Macrel is comparable to the best methods, with different trade-offs. In particular, Macrel achieves the highest precision and specificity at the cost of lower sensitivity. Although we do not possess good estimates of the proportion of AMPs in the smORFs predicted from real genomes (or metagenomes), we expect it to be much closer to 1:50 than to 1:3.

**Table 1 The comparison of Macrel AMP classifier performance and state-of-art methods shows that Macrel is among the best methods across a range of metrics.** The same test set (*Xiao et al., 2013*) was used to calculate the general performance statistics of the different classifiers, and the best value per column is in bold. Macrel refers to the Macrel classifier, while Macrel$^X$ is the same system trained with the *Xiao et al. (2013)* training set.

| Method | Acc. | Sp. | Sn. | Pr. | MCC | Reference |
|---|---|---|---|---|---|---|
| AmPEP* | **0.98** | – | – | – | **0.92** | *Bhadra et al. (2018)* |
| Macrel$^X$ | 0.95 | 0.97 | 0.94 | 0.97 | 0.91 | This study |
| iAMP-2L | 0.95 | 0.92 | 0.97 | 0.92 | 0.90 | *Xiao et al. (2013)* |
| Macrel | 0.95 | **0.998** | 0.90 | **0.998** | 0.90 | This study |
| AMAP | 0.92 | 0.86 | 0.98 | 0.88 | 0.85 | *Gull, Shamim & Minhas (2019)* |
| CAMPR3-NN | 0.80 | 0.71 | 0.89 | 0.75 | 0.61 | *Waghu et al. (2016)* |
| APSv2 | 0.78 | 0.57 | **0.99** | 0.70 | 0.61 | *Veltri, Kamath & Shehu (2018)* |
| CAMPR3-DA | 0.72 | 0.49 | 0.94 | 0.65 | 0.48 | *Waghu et al. (2016)* |
| CAMPR3-SVM | 0.68 | 0.40 | 0.95 | 0.61 | 0.42 | *Waghu et al. (2016)* |
| CAMPR3-RF | 0.65 | 0.34 | 0.96 | 0.59 | 0.39 | *Waghu et al. (2016)* |
| iAMPpred | 0.64 | 0.32 | 0.96 | 0.59 | 0.37 | *Meher et al. (2017)* |

**Notes:**
   * These data were retrieved from the original article.
   Acc, Accuracy; Sn, Sensitivity; Sp, Specificity; Pr, Precision; MCC, Matthew's Correlation Coefficient.

**Table 2 Macrel achieves accuracy comparable to the state-of-art in hemolytic peptides classification.** Models implemented by *Chaudhary et al. (2016)* were generically called HemoPI-1 due to the datasets used in the training and benchmarking (the best values per column are in bold).

| Method | Acc. | Sp. | Sn. | Pr. | MCC | Reference |
|---|---|---|---|---|---|---|
| HemoPI-1C,SVM* | **0.95** | 0.95 | **0.96** | 0.95 | **0.91** | *Chaudhary et al. (2016)* |
| HemoPI-1H* | **0.95** | 0.95 | **0.96** | 0.95 | **0.91** | *Chaudhary et al. (2016)* |
| HemoPI-1C,IBK* | **0.95** | 0.94 | **0.96** | 0.94 | 0.89 | *Chaudhary et al. (2016)* |
| HemoPI-1C,RF* | 0.94 | 0.95 | 0.94 | 0.95 | 0.89 | *Chaudhary et al. (2016)* |
| Macrel | 0.94 | **0.96** | 0.92 | **0.96** | 0.88 | This study |
| HemoPI-1C,Log* | 0.94 | 0.94 | 0.93 | 0.94 | 0.87 | *Chaudhary et al. (2016)* |
| HemoPI-1C,MP* | 0.93 | 0.93 | 0.94 | 0.93 | 0.87 | *Chaudhary et al. (2016)* |
| HemoPI-1C,JK48* | 0.89 | 0.88 | 0.90 | 0.89 | 0.78 | *Chaudhary et al. (2016)* |

**Note:**
   * These data were retrieved from the original article.

Therefore, we chose to use the higher precision classifier in Macrel for AMP prediction from real data to minimize the number of false positives in the overall pipeline.

Antimicrobial peptides, as they are likely to interact with cell membranes, can cause hemolysis, which can impact its potential uses, particularly in clinical settings (*Zhang & Gallo, 2016*; *Ruiz et al., 2014*; *Oddo & Hansen, 2017*). Therefore, for convenience, Macrel includes a classifier for hemolytic activity. This model is comparable to the state-of-the-art (see Table 2).

## High specificity is maintained when controlling for homology

Although we used out-of-bag estimates (see "Methods") to control for exact overlap between training and testing sets in the previous section, we still included *similar*

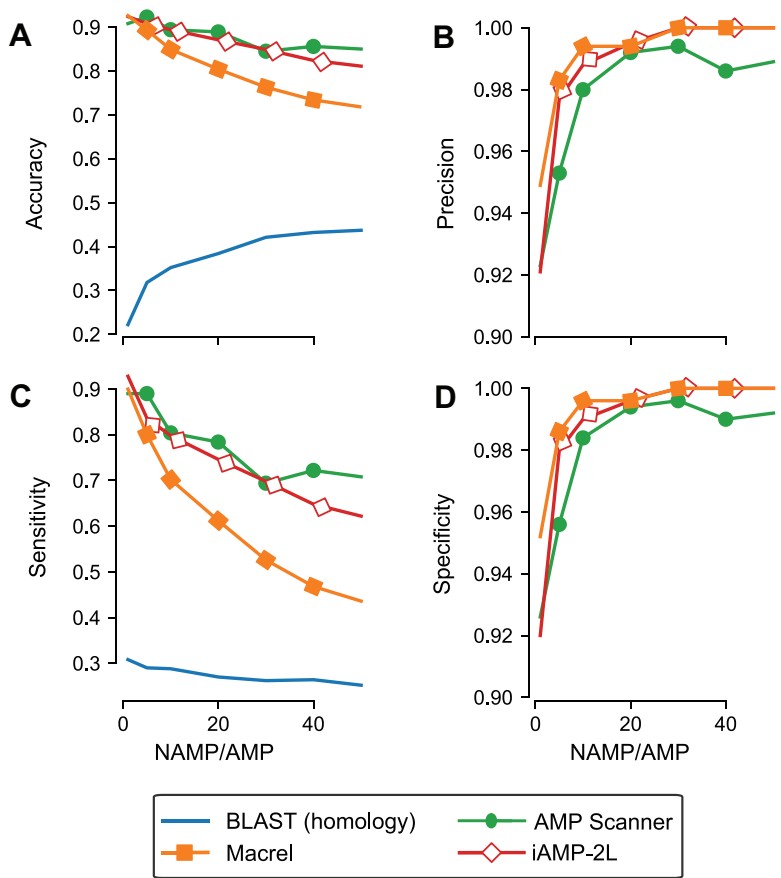

**Figure 3 Specificity is maintained even when controlling for homology in training and testing.** Different classifiers (blastp besthit as a purely homology-based system, Macrel, AMP Scanner v.2, and iAMP-2L) were trained with different proportions of non-AMPs:AMPs and tested on datasets which did not contain any homologous sequences in the training set (using an 80% identity cutoff). The results obtained in the homology-free datasets are showed in terms of accuracy (A), precision (B), Sensitivity (C), and Specificity (D).

sequences in training and testing, leading to an overestimate of generalization potential. To control for this effect, Macrel and three methods (those where the ability to retrain the model was provided by the original authors) were tested using a stricter, homology-aware, scheme where training and testing datasets do not contain homologous sequences between them (80% or higher amino acid identity, see "Methods").

As expected, the measured performance was lower in this setting, but Macrel still achieved perfect specificity. Furthermore, this specificity was robust to changes in the exact proportion of AMPs:non-AMPs used in the training set, past a threshold (see Table S4; Fig. 3D). Considering the overall performance of iAMP-2L model, future versions of Macrel could incorporate a combination of features from Macrel and iAMP-2L.

Using blastp as a classification method was no better than random, confirming that homology-based methods are not appropriate for this problem beyond very close homologs.

## Macrel recovers high-quality AMP candidates from genomes and metagenomes

To evaluate Macrel on real data, we ran it on 484 reference genomes that had previously shown to be abundant in the human gut (*Coelho et al., 2019*). This resulted in 171,645 (redundant) smORFs. However, only 8,202 (after redundancy removal) of these were classified as potential AMPs. Spurio (*Höps, Jeffryes & Bateman, 2018*) classified 853 of these (circa 10%) as likely spurious predictions.

Homology searches confirmed 13 AMP candidates as homologs from those in DRAMP database. Among them, a Laterosporulin (a bacteriocin from *Brevibacillus*), a BHT-B protein from *Streptococcus*, a Gonococcal growth inhibitor II from *Staphylococcus*, and other homologs of antimicrobial proteins. Seven of these confirmed AMPs were also present in the dataset used during model training.

To test Macrel on short reads, we simulated metagenomes composed of these same 484 reference genomes, at three different sequencing depths (40, 60, and 80 million reads) using abundance profiles estimated from six different real samples (*Coelho et al., 2019*) (for a total of 18 simulated metagenomes). The number of predicted smORFs increased with sequencing depth, with about 20k smORFs being predicted in the case of 80 million simulated reads (see Fig. 4A). Despite this large number of smORF candidates, only a small portion of them (0.17–0.64%) were classified as putative AMPs.

In total, we recovered 1,376 sequences for a total of 547 non-redundant AMPs predicted from the simulated metagenomes. Of these, only 44.5% are present in the underlying reference genomes. However, after eliminating singletons (sequences predicted in a single metagenome), this fraction rose to 80.4%. Thus, we recommend singleton elimination as a procedure to reduce false positives. Although fewer than half of pre-filtered AMP predictions were present in the reference genomes, only 12% of all predictions were marked as spurious by Spurio (see "Methods"). We manually investigated the origin of these spurious predictions and found that most of the spurious peptides are gene fragments from longer genes due to fragmentary assemblies or even artifacts of the simulated sequencing/assemblies. Interestingly, even the mispredictions were confirmed as AMPs by using the web servers of the methods tested in benchmark. In fact, circa 90% of all AMP candidates (including spurious predictions) were co-predicted by at least one other method than Macrel, and 61% were co-predicted by at least other four methods.

Having established that the rate of false positives can be kept low after singleton elimination, we investigated the recall of Macrel, namely whether it was able to recover the AMPs that were present in the underlying genomes. Post hoc, we estimated that almost all (97%) were in genomes with a coverage of at least 4.25 (while only 9% of the non-recovered AMPs had this, or a higher, coverage, see Fig. 4E). Nonetheless, in some exceptional cases, even very high coverage was not sufficient to recover a sequence.

## Macrel predicts putative AMPs in real human gut metagenomes

To evaluate Macrel on real data, we used 182 previously published human gut metagenomes (*Heintz-Buschart et al., 2016*). Of these, 177 (97%) contain putative AMPs, resulting in a total of 3,934 non-redundant sequences (see Table S3). The fraction of

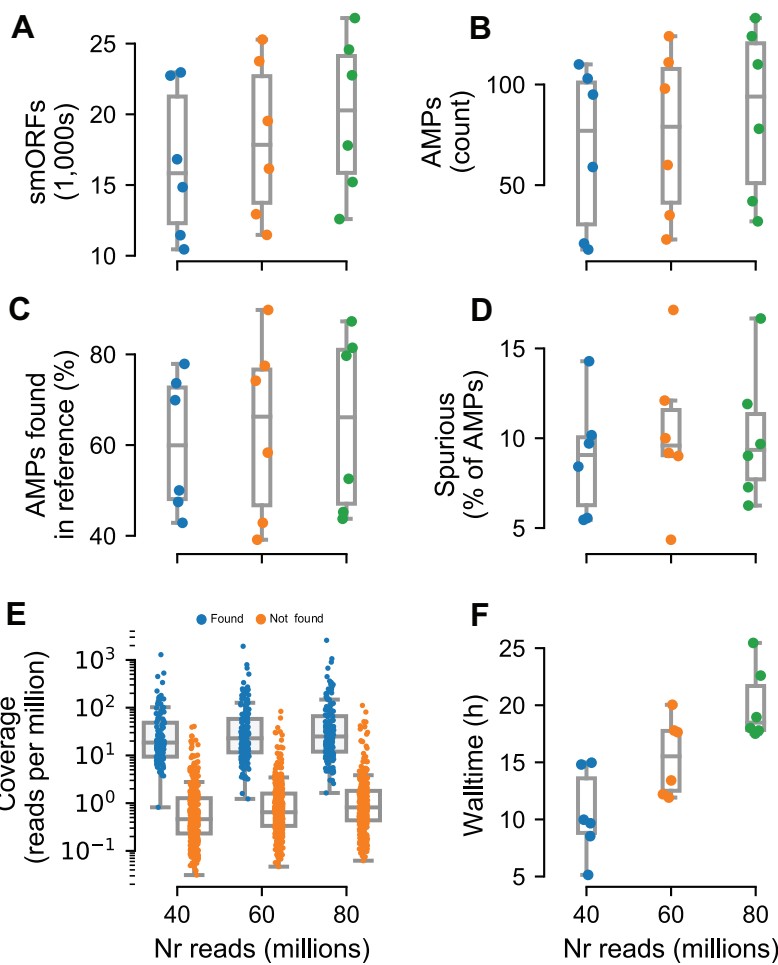

**Figure 4 Macrel results in six different metagenome simulations involving variation of the number of reads (40–80 million).** Six different microbial communities with realistic species abundances were simulated with increasing sequencing depth (see "Methods"). Macrel recovers a large number of small ORFs (smORFs) per metagenome (A), and a small number of AMPs from each metagenome (B). The number of AMPs returned that were present in the reference genomes covers a large range (40–90%) (C), but only a small fraction is detected as being a spurious prediction (D) (see "Methods"). Detection of AMPs is heavily dependent on coverage (E), with almost all (97%) of the detected AMPs contained in genomes with coverage above 4.25 (this is the simulated coverage of the genome, which, due to the stochastic nature of the process, will only correspond to the local coverage, on average). Processing times increase with coverage, with the single largest sample taking 25.5 h (F). In all panels, boxplot whiskers represent 1.5 times the inter-quartile range (capped to the 0–100% range where appropriate).

smORFs classified as AMPs per metagenome ranged 0.1–1.65%, a range similar to that observed in simulated metagenomes (see "Macrel Recovers High-Quality AMP Candidates from Genomes and Metagenomes").

After eliminating singletons, 1,373 non-redundant AMP candidates remained, which we further tested with alternative methods. In total, 92.8% of the AMPs predicted with Macrel were also classified as such by at least one other classifier, and 65.5% of the times, half or more of the tested state-of-art methods agreed with Macrel results (see Table S5).

iAMPpred and CAMPR3-RF showed the highest agreement and co-predict 74.4% and 65.7% of the AMPs predicted by Macrel, respectively.

Ten percent of all predicted AMPs (414 peptides, or 10.5%) were flagged as likely spurious (see "Methods"). The fraction of non-singleton AMPs predicted as spurious was slightly lower (8%, a non-significant difference). Our final dataset, after discarding both singletons and smORFs identified as spurious (see "Methods" and Table S3), consists of 1,263 non-redundant AMPs.

As 36 metatranscriptomes produced from the same biological samples are also available, we quantified the expression of the 1,263 AMP candidates. Over 53.8% of the predicted AMPs had detectable transcripts (see Fig. S2). For 72% of these, transcripts were detected in more than one metagenome.

Taken together, we concluded that Macrel could find a set of high-quality AMPs candidates, which extensively agrees with other state-of-art methods, many of which are being actively transcribed.

### Macrel requires only moderate computational resources

The tests reported here were carried out on machines corresponding to standard consumer hardware (Amazon WebServices, t2.large, which feature 8 GB of RAM and 2 cores) to show that Macrel is a pipeline with modest computational requirements. The execution time, although naturally dependent on the input size, was not greater than 25.5 h (recall that the largest simulated metagenomes contained 80 million reads, see Fig. 4F). The assembly steps consumed 75–80% of the execution time, while read trimming and gene prediction occupied another considerable part (10–15%).

## DISCUSSION

Using a combination of local and global sequence encoding techniques, Macrel classifiers perform comparably to the state-of-the-art in benchmark datasets. These benchmarks are valuable for method development, but as they contain the same number of AMP and non-AMP sequences in the testing set, are not a good proxy for the setting in which we intend to use the classifiers. It is unlikely that half of peptide sequences predicted from genomes and metagenomes will have antimicrobial activity. Therefore, we chose a classifier that achieves a slightly lower accuracy on these benchmarks, but has very high specificity.

We also presented an initial analysis of publicly-available human gut metagenomes (Heintz-Buschart et al., 2016). The 1,263 AMPs predicted with Macrel were largely congruent (92.8%) with other state-of-art methods. This opens up the possibility of future work to understand the impact of these molecules on the microbial ecosystems or prospecting them for clinical or industrial applications.

Some AMPs are the result of post-translational modifications (Ortega & Van der Donk, 2016; Arnison et al., 2013; Agrawal et al., 2017). In version 1.0, however, Macrel only extracts AMPs that are present in the genome (or metagenome) encoded in their active form. This is the classification supported by the other tools in the comparison, although very recently, Fingerhut et al. (2020) presented ampir, which does support detection of precursor sequences. Future releases will extend Macrel in that direction.

## CONCLUSIONS

Macrel performs all operations from raw metagenomic read assembly to the prediction of AMPs. The main challenge in computationally predicting smORFs (small ORFs, such as AMPs) with standard methods is the high rate of false-positives. However, after the filtering applied by Macrel classifiers, only a small number of candidate sequences remained. Supported by several lines of evidence (low level of detected spurious origin, similar classification by other methods, and evidence of AMPs transcription), we conclude that Macrel produces a set of high-quality AMP candidates.

Macrel is available as open-source software at https://github.com/BigDataBiology/macrel and the functionality is also available as a webserver: https://big-data-biology.org/software/macrel.

## ACKNOWLEDGEMENTS

We thank Hiram He, Fudan University, who helped set up the Macrel website and kindly offered coding support as well as members of the Coelho group for helpful comments on previous versions of the manuscript. We thank beta users of Macrel for their comments and bug reports.

### Funding

This work was supported by the National Key R&D Program of China (2020YFA0712403, 2018YFC0910500), the National Natural Science Foundation of China (61932008, 61772368), the Shanghai Science and Technology Innovation Fund (19511101404 and the Shanghai Municipal Science and Technology Major Project (2018SHZDZX01). There was no additional external funding received for this study. The funders had no role in study design, data collection and analysis, decision to publish, or preparation of the manuscript.

### Grant Disclosures

The following grant information was disclosed by the authors:
National Key R&D Program of China: 2020YFA0712403, 2018YFC0910500.
National Natural Science Foundation of China: 61932008, 61772368.
Shanghai Science and Technology Innovation Fund: 19511101404.
Shanghai Municipal Science and Technology Major Project: 2018SHZDZX01.

### Competing Interests

The authors declare that they have no competing interests.

### Author Contributions

- Célio Dias Santos-Júnior conceived and designed the experiments, performed the experiments, analyzed the data, prepared figures and/or tables, authored or reviewed drafts of the paper, and approved the final draft.

- Shaojun Pan performed the experiments, prepared figures and/or tables, authored or reviewed drafts of the paper, and approved the final draft.
- Xing-Ming Zhao conceived and designed the experiments, authored or reviewed drafts of the paper, and approved the final draft.
- Luis Pedro Coelho conceived and designed the experiments, performed the experiments, analyzed the data, prepared figures and/or tables, authored or reviewed drafts of the paper, and approved the final draft.

## Data Availability

Code for Macrel was archived and is available at Zenodo: Célio D. Santos-Júnior Luis Pedro Coelho, Hiram He, & psj1997. (2020, October 28). BigDataBiology/macrel: Version 0.6.1 (Version v0.6.1). Zenodo. DOI 10.5281/zenodo.4147382.

It is also available on GitHub: https://github.com/BigDataBiology/macrel

Code for the benchmarks is available at GitHub: https://github.com/BigDataBiology/macrel2020benchmark.

Newly simulated data was similarly archived and is available at Zenodo:

Santos-Júnior, Célio Dias, Pan, Shaojun, Zhao, Xing-Ming, & Coelho, Luis Pedro. (2019). Macrel software benchmark data set: Simulated metagenomes with sequencing quality, errors profile and abundance distributions derived from real samples (Version v.1.0) [Data set]. Zenodo. DOI 10.5281/zenodo.3529860.

## Supplemental Information

Supplemental information for this article can be found online at http://dx.doi.org/10.7717/peerj.10555#supplemental-information.

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
