# Peer review of "Macrel: antimicrobial peptide screening in genomes and metagenomes"

_PeerJ, doi:10.7717/peerj.10555_

## Round 0.1 · original submission · Minor Revisions

Both reviewers were very positive about the state of the manuscript, but I agreed particularly with reviewer 1 that there were a couple of places that this could be improved for those readers in adjacent fields.

Reviewer 1 ·

Basic reporting

Overall, this is a very high quality submission, with professional English used throughout. It is extremely well cited, which was important given the dizzying array of software methods used. The article is structured appropriately, and the figures are clear.

I noticed 2 typographical errors (line 56 "Hidrophobicity" and line 174 "biased amino compositions").

I do feel the article is not self-contained. The authors make an extensive set of comparisons of their tool to many others, with little explanation of the state of the field. While I think the content is well written for an expert in this subfield, I would strongly recommend devoting more space in the introductory section to bringing other secondary metabolism/natural products researchers up to speed. Perhaps at least brief explanations of the current software approaches that you'll compare against in the Results, and their strengths and weaknesses. What makes Macrel unique? Given that the actual results are not miles better than existing tools, be sure to explain why Macrel holds an important niche here (as I think it does).

Experimental design

The experimental design is excellent, and fits well within the Aims and Scope of PeerJ. The investigation is rigorous, and there were no detectable violations of standards or leading in the author's experimental work. I applaud the authors' efforts to make all data available, all code and scripts available, and clear methodology that should allow for easy reproducibility.

The only issue I had was that the manuscript does not make clear to me why this machine learning method seems to be exclusively used on metagenome sequence, especially as the metagenome sequence is first trimmed and assembled. Is there a reason why it will not or cannot work well on isolate genomes or isolate genome reads? Prediction of AMPs in well-established model genomes would provide a better avenue for confirming results biochemically. (I am not suggesting that chemical confirmatory experiments be done for this manuscript, but I do think the natural products community would be interested in this.) If there is an obvious explanation that I'm missing, I feel it would be best to articulate it more clearly in the intro and the Conclusions.

Validity of the findings

The findings are well articulated and valid. These are not high-impact results, but that is not a criteria for PeerJ, and the honest appraisal of the method is well received.

Additional comments

Overall, I think this is a well-prepared manuscript, but could perhaps be improved with the changes listed above, and I think it should be accepted pending minor revisions. These suggested revisions are:
- A bit more explanation on the state of AMP prediction (ie description of specific tools compared against in the Results section)
- An explanation of why Macrel is used on metagenomic sequence, and not isolate genomes
- Very minor typographical errors

·

Basic reporting

no comment

Experimental design

no comment

Validity of the findings

I played around with a number of lanthipeptide sequences that I had on hand, and it seems to me that they all achieve higher AMP probabilities once I manually remove the leader peptide part. This seems to match the example of microbisporicin A1 the authors use in figure 2 c, where the coordinates also seem to be for the processed lanthipeptide, not the prepropeptide. Can the authors comment a bit how this would impact the ability of MACREL to pick up on RiPPs, which usually contain a leader or tail peptide?

Additional comments

In their manuscript "MACREL: antimicrobial peptide screening in genomes and metagenomes", Dias Santos-Junior et al. describe a software pipeline to screen for antimicrobial peptides in (meta)genome data. The manuscript is well-written, the source code and training/evaluation datasets are available. Using the provided install script, the tool is very easy to install via conda. This is a nice break from many other bioinformatics tools.

The tool is straightforward to use, and comes with a user manual on ReadTheDocs. A minor comment I'd have on the user interface is that it'd be nice if --help would actually list the commands available.

---

## Round 0.2 · accepted · Accept

Both reviewers were satisfied that all points had been addressed, so we're happy to accept this for publication.

Reviewer 1 ·

Basic reporting

As this is a second review, I won't go through the checklist again. No further issues.

Experimental design

No further issues

Validity of the findings

No further issues

Additional comments

I thank the authors' for their thoughtful responses to my first review, and the adjustments made to the manuscript are well-considered and quite acceptable. I recommend publication in its current form.

·

Basic reporting

no comment

Experimental design

no comment

Validity of the findings

no comment

Additional comments

The authors have responded to my questions to my full satisfaction, thanks!